# Differences in lower-limb biomechanics during single-leg landing considering two peripheral fatigue tasks

**Makoto Asaeda** [1,2]*, **Kazuhiko Hirata**[2], **Tomoya Ohnishi**[1], **Hideyuki Ito**[1], **So Miyahara**[1], **Yukio Mikami**[3]

**1** Faculty of Wakayama Health Care Sciences, Takarazuka University of Medical and Health, Wakayama, Japan, **2** Division of Rehabilitation, Department of Clinical Practice and Support, Hiroshima University Hospital, Hiroshima, Japan, **3** Department of Rehabilitation Medicine, Hiroshima University Hospital, Hiroshima, Japan

* asaeda.m@gmail.com

**Data Availability Statement:** All relevant data are within the manuscript.

**Funding:** The author(s) received no specific funding for this work.

## Abstract

Dynamic knee valgus (DKV) occurs during landing after a fatigue task involving the lower extremity. However, the manner in which different peripheral fatigue tasks affect DKV remains unknown. In this study, we investigated the DKV via electromyography during single-leg landing considering the hip-joint fatigue task (HFT) and knee-joint fatigue task (KFT) performed by healthy men. We recruited 16 healthy male participants who performed a single-leg jump-landing motion from a height of 20 cm before and after an isokinetic hip abduction/adduction task (HFT) and knee extension/flexion task (KFT). Three-dimensional motion analysis systems were attached to the left gluteus medius and quadriceps, and surface electromyography was used to analyze the lower limb kinematics, kinetics, and muscle activity. The primary effects and interactions of the task and fatigue were identified based on the two-way repeated-measures analysis of variance. The results of the average angle during landing indicated that DKV occurs in KFT, whereas HFT applies external forces that adduct and internally rotate the knee at peak vertical ground reaction force (vGRF). Furthermore, both KFT and HFT exhibited an increase in muscle activity in the quadriceps. The analysis revealed that the occurrence of DKV varies depending on the peripheral fatigue task, and the effects on average DKV during landing and DKV at peak vGRF vary depending on the peripheral fatigue task.

## Introduction

Anterior cruciate ligament (ACL) injury is a common sports injury that occurs in people leading an active lifestyle. ACL injuries are caused by the stress applied to the ligaments because of tibial internal rotation and posterior displacement associated with knee valgus [1, 2]. The consensus for preventing an ACL injury involves suppressing the dynamic knee valgus (DKV) motion. However, a meta-analysis reports that the indicators of DKV, namely the knee valgus

**Competing interests:** The authors have declared that no competing interests exist.

angle and valgus moment, calculated using a three-dimensional (3D) motion analysis device cannot predict ACL injuries [3].

Benjaminse et al. [4] reported that considering multiple factors is crucial for understanding the DKV dynamics, and muscle weakness in the hip joint is an effect of DKV. They examined certain peripheral fatigue tasks that induced DKV, focusing on the hip and knee joints involved in posture control during jump-landing. The physiological processes that can cause fatigue are classically divided into two domains: the activation level of the muscle (central) and the other influences on contractile function (peripheral) [5]. Fatigue is also classified as perceived or performance fatigability, and it is determined by a wide variety of factors. Peripheral fatigue includes neuromuscular function, metabolism, contractile apparatus, and contraction coupling in peripheral muscles, which is related to fatigue as a task dependency [5, 6]. A previous study used electromyography (EMG) and reported that the hip abduction moment in healthy females is reduced by 27% owing to the weakening of the gluteus medius [7]. Fatigue in the quadriceps and hamstrings is known to significantly increase the knee extension and external rotation at the peak vertical ground reaction force (vGRF) [8]. However, systematic reviews have reported that kinematics and kinetics data from the fatigue tasks are not consistent. Moreover, only a few studies have investigated the biomechanics of lower limbs considering different tasks [9]. Xia et al. reported that no significant differences exist in terms of peak vGRF between constant running fatigue protocol and the combination of shuttle running and vertical jumping [10]. The comparison of the slow linear fatigue protocol and short-term fatigue protocol indicates that the internal adduction moment is greater during the latter [11]. However, the differences in lower-limb biomechanics during single-leg landing based on two peripheral fatigue tasks are not yet explored. Previously, clinical trials have shown that hip-focused training can prevent ACL injuries [12]; therefore, it is necessary to examine not only the general fatigue protocol, such as that reported in previous studies, but also the differences observed in the peripheral fatigue protocol by focusing on the joints of the lower limbs.

The objective of this study is to clarify the differences observed during DKV in terms of the frontal plane hip-joint and knee-joint angle/moment and perform EMG considering two peripheral fatigue tasks, namely, the knee-joint fatigue task (KFT) and hip-joint fatigue task (HFT). Additionally, we clarify the relationship between the degree of peripheral muscle fatigue and the changes in DKV based on EMG. We further examine how peripheral muscle fatigue causes DKV. We hypothesized that DKV increases after both tasks, the activity in the gluteus medius decreases after HFT, and the quadriceps muscle activity decreases after KFT. Furthermore, we determined that DKV is more likely to occur when the decline in EMG and muscle strength is high.

## Methods

### Participants

This was a single-case study with a prospective single group, and we performed a repeated-measures study in a single campus. Several healthy adult males volunteered to participate in the research after viewing a poster in a public space (at the campus entrance), and participants were recruited from April 2021 to September 2022. We recruited those (i) who could understand the study consent in writing; (ii) without a history of orthopedic surgery in the lower limb neuromuscular disease; (iii) without pain; (iv) exhibiting a limited range of motion in the lower limb, which affected their jump-landing; and (v) carrying out regular sports activities. A questionnaire was administered to prospective participants to confirm their history of orthopedic surgery related to lower-limb neuromuscular diseases and to determine the presence or absence of pain. Additionally, joint angles were measured using a goniometer to assess

individuals with limited range of motion in the lower limb. The exclusion criteria included individuals with a history of orthopedic surgery or lower-limb pain and those with limited joint range of motion (i.e., less than 135° of flexion), maintaining the posture when jumping and landing, and not carrying out regular sporting activities, carrying out sporting activities at competition level (e.g. professional sports level). Heights and weights of all participants were respectively measured using height scale (Yamato, Japan) and weight scale (Yamato, Japan), and BMI was calculated. This study was performed with the approval of the Institutional Research Ethics Committee and in accordance with the Declaration of Helsinki (obtained with informed consent).

The number of participants was selected based on a previous study that reported a significant increase in the internal knee rotation moment in males after a fatigue task was performed (until 50% of the baseline peak torque of knee extension and flexion) to assume the occurrence of DKV [8]. We calculated the sample size considering a significance level of 5%, power = 0.8, and $p < 0.05$ at G*power (version 3.1.9.4); the program was written, conceptualized, and designed by Franz, University of Kiel, Germany. The required sample size was determined to be 16.

## Landing task

The participants wore clothes with high body adhesion, and 39 infrared reflective markers, with a diameter of 10 mm, were attached to the entire body per the Plug-in Gait model of the Vicon Motion System. Data were collected in a secured room, ensuring minimal external influence during measurements, and the measurements were conducted by two individuals: MA and HI.

The one-leg jump landing movement was performed from a height of 20 cm; this was based on a previous study [13]. The upper limb was lowered, the right hip was in an intermediate position, and the knee joint was slightly flexed in a one-leg standing posture from which the subject "fell" from a height of 20 cm. This movement was performed before and immediately after the fatigue task. Furthermore, participants were requested to practice the movement three times before executing the fatigue task (pre-fatigue). We adopted a duration of 5 s to maintain a successful landing attitude; the participants performed three successful landings. Before the fatigue task, participants practiced the landing task three times, but no practice was deemed necessary for the movement immediately after the fatigue task (post-fatigue). All participants had a history of competing in sports but lacked prior experience in performing the specific jump landing motion used in this study.

During the jump-landing measurement, the data on the marker positions and the GRF data were collected using a 3D motion analysis device (Vicon MX, Vicon Motion Systems, UK) and two force plates (AMTI, USA). The camera frequency for the 3D motion analysis was set to 100 Hz, and the GRF for each force plate was recorded at 1000 Hz. The obtained marker position data were passed through a 6-Hz fourth-order Butterworth low-pass filter.

## Fatigue tasks

The isokinetic movement was adopted as the fatigue task. The participants performed the isokinetic knee extension/flexion movement (KFT) and hip abduction/adduction movement (HFT) as per the Biodex Medical Systems (USA). The muscle torque values were recorded for both tasks during the first and final three contractions. The degree of subjective fatigue was evaluated using the Borg scale.

The KFT was performed in a sitting position with the chair tilted at 8°; the participant held the bars of the chair with both upper limbs. The attachment was placed distal to the lower leg

(2 cm proximal to the medial malleolus), and the axis of rotation was aligned with the line connecting the medial and lateral condyles of the femur. Additionally, gravity correction was performed at a knee flexion of 20°. The participants performed a knee-joint extension movement with maximum effort, starting from the 135° knee-joint flexion. Subsequently, the flexion was performed with maximum effort when reaching the extension of 0°. The angular velocity was set to 120°/s, and the movements were performed 40 times each; these were set based on the reports of a study confirming adequate reliability of the test [14]. Fig 1A depicts the KFT results.

A

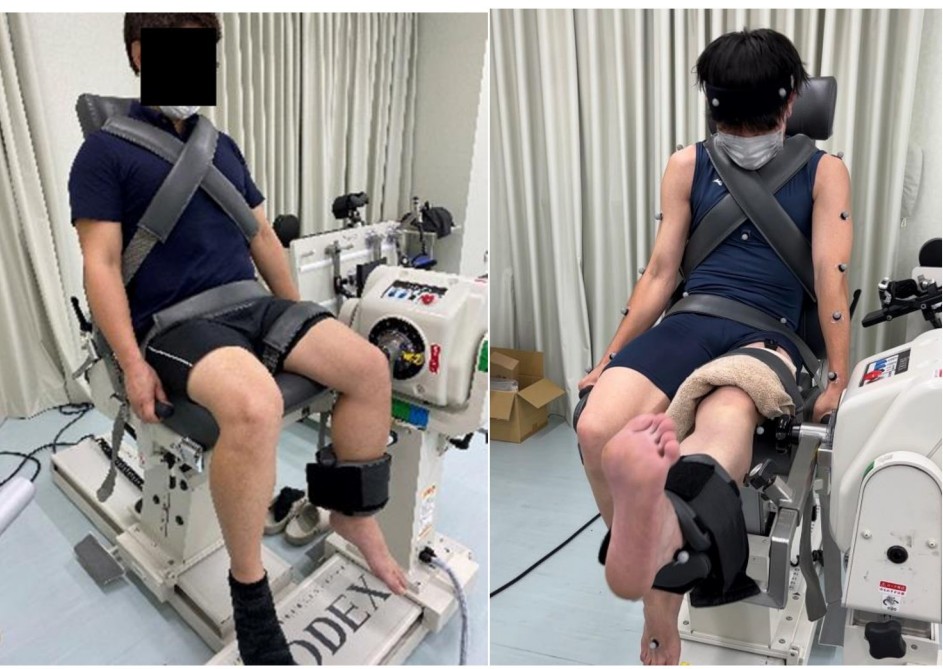

B

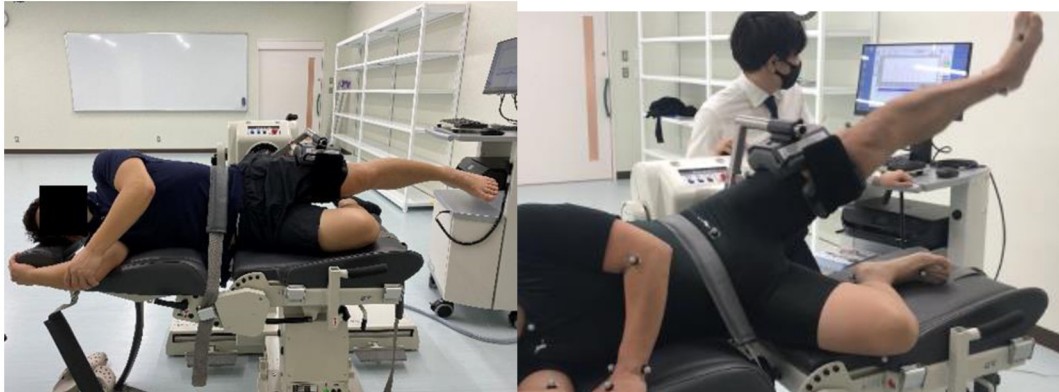

**Fig 1. Positions during the fatigue task.** (A) Knee-joint fatigue task (KFT): left.; starting position, right; during task (B) Hip-joint fatigue task (HFT): left.; starting position, right; during task.

The HFT was performed on the left side, with the right hip and knee joints slightly flexed. The participant then held the head of the chair and grasped the bar using both upper limbs. The attachment was placed on the distal thigh (proximal knee joint), and the axis of rotation was aligned with the upper part of the greater trochanter. The gravity correction of lower limbs was performed at 30˚ hip abduction. Furthermore, the range of movement was set to 10˚ hip adduction and 30˚ abduction, and the angular velocity was set to 60˚/s for each measurement method; this exhibited adequate reliability [15, 16]. Each abduction/adduction was performed 40 times. Fig 1B illustrates an image of the HFT.

## Data analysis

Data analysis was performed at the data collection site by a single individual using a computer having no internet connectivity. The analyzed results were reviewed by two individuals, TO and SM, to ensure their validity. During DKV, the angles and moments of the hip and knee joints were calculated from the obtained marker position and GRF data using the Plug-in Gait model. The time point at which vGRF exhibited 10 N was defined as the initial contact (IC). We calculated the average DKV at 100 ms from the IC and the DKV at the peak vGRF.

The EMG measurement was performed by attaching the Trigno Avanti Sensor Wireless System (Delsys Inc., USA) to the left gluteus medius and quadriceps according to a previous study [17]. The sampling frequency was 2000 Hz, and the bandwidth was 20–450 Hz. Based on a previous study [18], we defined IC as the point at which vertical acceleration is observed immediately before the built-in acceleration sensor indicates the maximum downward acceleration. The analysis interval was set to 100 ms from IC, and in this analysis interval during EMG, we performed offset removal, root mean square calculation (window length: 0.1, window laps: 0.09), and division of maximum voluntary isometric contraction. The maximum voluntary isometric contraction of gluteus medius was performed with the participants maintaining a 10˚ posture of hip abduction in the side position, whereas that of the quadriceps was obtained with the participants performing a 90˚ flexion of the hip and knee in the sitting position. The peak amplitude was the maximum value determined during the first 100 ms starting from IC.

## Statistical analysis

Statistical analysis was performed using IBM SPSS Statistics 27 (IBM Japan, Japan). We used the two-way repeated-measures analysis of variance to identify the primary effects and interactions of the task and fatigue. After identifying significant interactions, the simple main effects were evaluated using Bonferroni correction. The statistical significance was set to $p < 0.05$, and all analyses were performed using SPSS.

## Results

Overall, 21 individuals volunteered to participate in the study. Among them, we separated those who had an ankle fracture in the previous year (n = 1) from the questionnaire before the landing task or experienced pain during landing in the practice phase before the fatigue task (n = 1). Additionally, we excluded one participant who had difficulty maintaining the posture when jumping and landing and two participants whose EMG data were lost during the fatigue task because of dropping the electrodes for surface electromyography during the fatigue task; therefore, 16 participants were included in the analysis. Table 1 summarizes the characteristics of the participants and the peak torque observed during the fatigue task. Both KFT and HFT exhibited a significant decrease in peak torque, whereas the Borg scale showed no significant

**Table 1. Characteristics of the participants.**

| | | Mean | SD | | |
|---|---|---|---|---|---|
| Age (year) | | 19.8 | 0.9 | | |
| Height (cm) | | 173.5 | 7.2 | | |
| Body weight (kg) | | 66.4 | 11.7 | | |
| BMI | | 21.9 | 2.7 | | |
| Peak torque during fatigue task (% body weight) | | Knee extension (KFT) | | Hip abduction (HFT) | |
| | | Mean | SD | Mean | SD |
| | First 3 times | 2.141 | 0.289 | 1.525 | 0.366 |
| | Last 3 times | 1.005* | 0.246 | 0.878* | 0.271 |
| Borg scale after fatigue task | | Median | IQR | Median | IQR |
| | | 17.0 | 16.0–18.0 | 16.5 | 13.5–17.0 |

KFT: knee fatigue task; HFT: hip fatigue task; IQR: Interquartile range; BMI: body mass index; SD: standard deviation

*: significantly decreased compared with the value of the first three times ($p < .001$)

difference. All participants presented recreational levels in terms of sports activity, and none performed at the competitive level.

In terms of the average DKV, all except the hip internal rotation moment exhibited interaction. In KFT, the hip adduction angle and moment decreased (Fig 2A and 2C, $p = .002$ and $< .001$); the hip internal rotation angle decreased (Fig 2B, $p < .001$); the knee adduction angle and moment decreased (Fig 3A and 3C, both $p < .001$); the knee internal and external rotation angles increased and decreased, respectively (Fig 3B, $p < .001$); and the knee internal rotation moment decreased (Fig 3D, $p = .003$). In the post-fatigue task, the hip adduction moment was

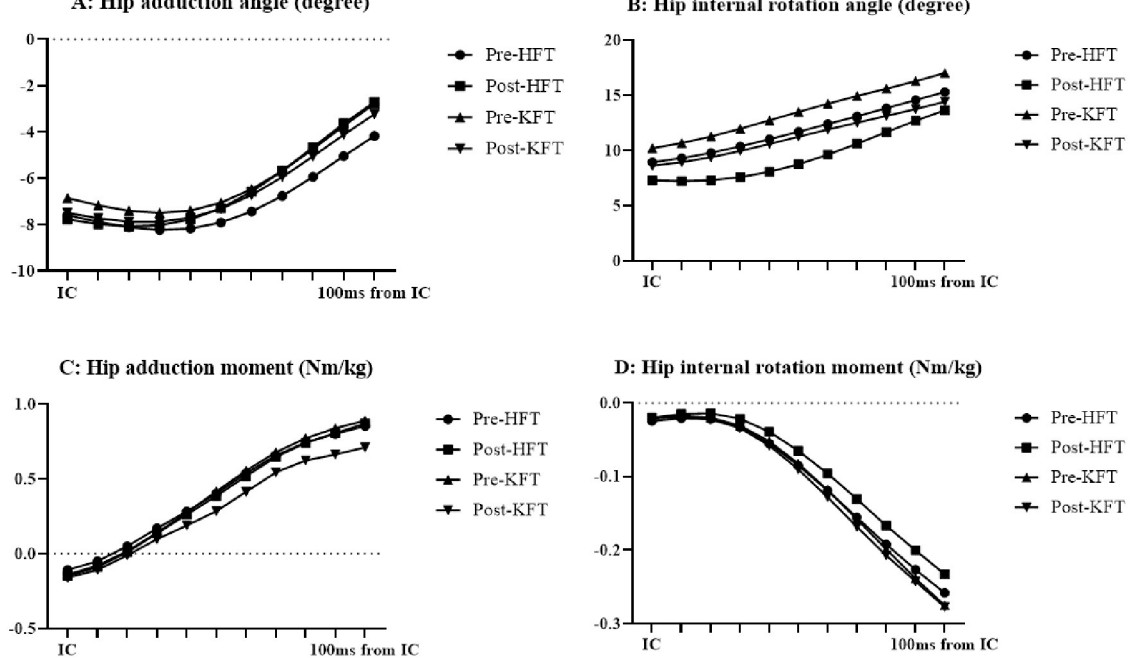

**Fig 2. Time course of hip kinematics and kinetics (average DKV).** (A) Hip adduction angle. (B) Hip internal rotation angle. (C) Hip adduction moment. (D) Hip internal rotation moment. DKV: dynamic knee valgus; HFT: hip-joint fatigue task; IC: initial contact; KFT: knee-joint fatigue task.

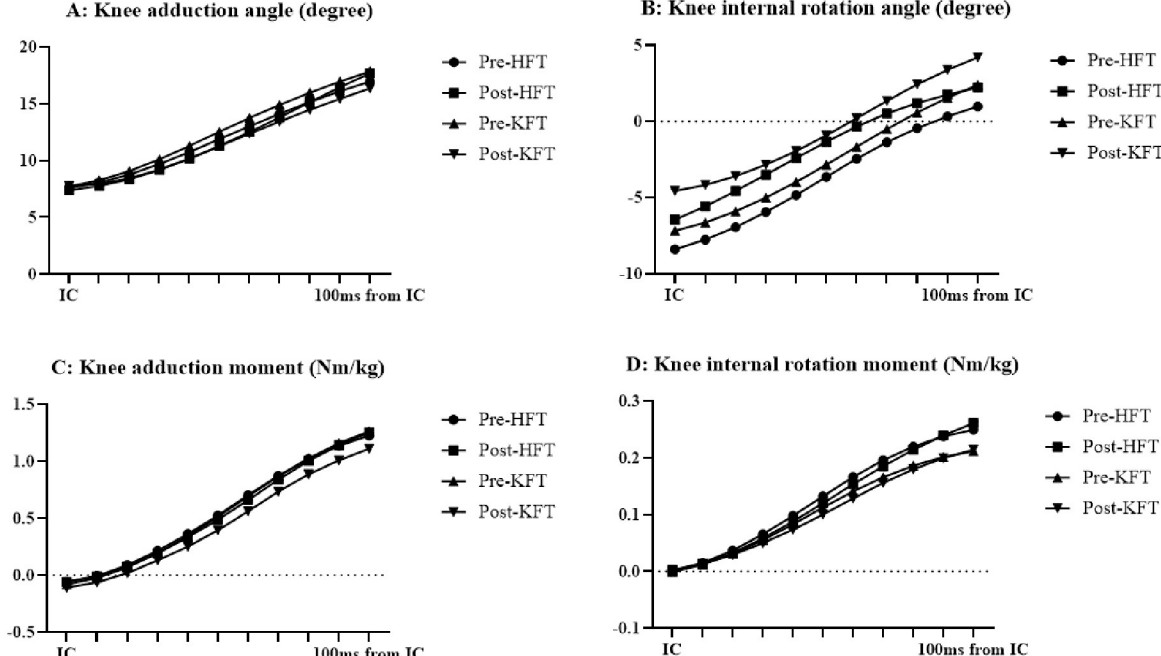

**Fig 3. Time course of knee kinematics and kinetics (average DKV).** (A) Knee adduction angle. (B) Knee internal rotation angle. (C) Knee adduction moment. (D) Knee internal rotation moment. DKV: dynamic knee valgus; HFT: hip-joint fatigue task; IC: initial contact; KFT: knee-joint fatigue task.

lower (Fig 2B, $p = .047$), whereas the knee internal rotation angle was higher in the KFT than in HFT (Fig 3B, $p = .003$).

DKV at peak vGRF exhibited significant interaction at the hip adduction moment, knee adduction angle/moment, and knee internal rotation moment (Table 2). The knee adduction/internal rotation moment increased after performing the fatigue task in HFT (Table 3).

**Table 2. ANOVA in dynamic knee valgus at peak vGRF.**

| | | | ANOVA (*F* value; *F* [1, 30]) | | |
|---|---|---|---|---|---|
| | | | **Time** | **Task** | **Interactions** |
| Hip | Angle | | | | |
| | | Adduction | 0.591 | 0.367 | 0.379 |
| | | Internal rotation | 0.323 | 0.373 | 0.011 |
| | Moment | | | | |
| | | Adduction | 0.030 | 1.369 | 8.753* |
| | | Internal rotation | 2.104 | 0.274 | 0.087 |
| Knee | Angle | | | | |
| | | Adduction | 0.970 | 0.027 | 4.913* |
| | | Internal rotation | 2.662 | 0.481 | 0.360 |
| | Moment | | | | |
| | | Adduction | 1.094 | 0.118 | 6.170* |
| | | Internal rotation | 8.577* | 2.545 | 5.833* |

ANOVA: analysis of variance; vGRF: vertical ground reaction force;

*: $p < .050$ by two-way repeated measures ANOVA

**Table 3. Dynamic knee valgus at peak vGRF.**

| Time | | Pre-fatigue | | | | Post-fatigue | | | |
|---|---|---|---|---|---|---|---|---|---|
| Task | | KFT | | HFT | | KFT | | HFT | |
| | | Mean | SD | Mean | SD | Mean | SD | Mean | SD |
| Hip | Angle | | | | | | | | |
| | Adduction | 0.087 | 5.615 | -1.307 | 5.484 | 0.153 | 4.912 | -0.709 | 5.596 |
| | Internal rotation | 20.00 | 10.47 | 18.03 | 9.528 | 18.19 | 13.07 | 16.20 | 10.13 |
| | Moment | | | | | | | | |
| | Adduction | 0.964 | 0.282 | 0.976 | 0.386 | 0.849 | 0.335 | 1.105 | 0.368 |
| | Internal rotation | -0.352 | 0.125 | -0.339 | 0.088 | -0.375 | 0.125 | -0.355 | 0.088 |
| Knee | Angle | | | | | | | | |
| | Adduction | 20.47 | 7.433 | 19.29 | 8.480 | 19.54 | 9.948 | 21.70 | 9.150 |
| | Internal rotation | 4.186 | 9.978 | 2.377 | 11.39 | 6.281 | 6.840 | 3.345 | 11.25 |
| | Moment | | | | | | | | |
| | Adduction | 1.915 | 0.371 | 1.246* | 0.458 | 1.249 | 0.340 | 1.409 | 0.454 |
| | Internal rotation | 0.191 | 0.105 | 0.226* | 0.113 | 0.198 | 0.129 | 0.297 | 0.145 |

KFT: knee fatigue task; HFT: hip fatigue task; vGRF: vertical ground reaction force; SD: standard deviation;

*: $p < .050$ by Bonferroni correction

Furthermore, no significant simple main effect was observed for the hip adduction moment or the knee adduction angle (Table 2).

Although no interaction was observed in the EMG comparison, a main effect of time occurred at the peak amplitude of the quadriceps (Table 4). However, no significant main effects or interactions existed in the gluteus medius (Table 4).

## Discussion

The objective of this study was to clarify the differences in the lower-limb biomechanics during DKV with respect to the hip-joint and knee-joint angles/moments in the frontal and transverse planes based on the EMG performed for KFT and HFT. Accordingly, the average angle during landing indicated that DKV occurs in KFT, and HFT applies external forces that adduct and internally rotate the knee at peak vGRF. Furthermore, both KFT and HFT exhibited an increase in quadriceps muscle activity. Our analysis revealed that the occurrence of DKV, the effects on average DKV during landing, and DKV at peak vGRF vary depending on the peripheral fatigue task.

**Table 4. Electromyography during landing.**

| | Pre-fatigue | | Post-fatigue | | ANOVA ($F$ value; $F$ [1, 30]) | | |
|---|---|---|---|---|---|---|---|
| | Mean | SD | Mean | SD | Time | Task | Interactions |
| Gluteus medius | | | | | | | |
| KFT | 62.41 | 28.31 | 68.73 | 45.63 | 1.188 | 0.264 | 0.002 |
| HFT | 57.50 | 22.66 | 63.63 | 28.23 | | | |
| Quadriceps | | | | | | | |
| KFT | 46.66 | 16.78 | 59.21 | 28.47 | 15.59* | < 0.001 | 0.265 |
| HFT | 54.19 | 30.85 | 70.96 | 59.80 | | | |

Unit: %MVIC; KFT: knee fatigue task; HFT: hip fatigue task; MVIC: maximum voluntary contraction quantified by isometric contraction; SD: standard deviation;

ANOVA: analysis of variance;

*: $p < .050$ by two-way repeated measures analysis of variance (ANOVA)

In terms of the peripheral fatigue of the hip, a previous study reported that the external hip and knee adduction moment exhibited a decrease of 24% after an isometric hip adduction task [7]. In this study, the external hip adduction moment did not significantly change after the iso-kinetic hip abduction/adduction; however, the knee adduction/internal rotation moments were significantly higher after HFT. Furthermore, the EMG results indicated no significant changes in gluteus medius activity, whereas the quadriceps muscle activity was increased. Therefore, no difficulty was observed in terms of postural control that could result in DKV, which in turn would have been compensated for by quadriceps muscle activity. The lack of change in gluteus medius muscle activity supports the fact that no significant difference was observed in the hip adduction moment. In previous studies, peak knee abduction moments have been negatively correlated with quadriceps activity during landing tasks [19]. In this study, quadriceps activity may have influenced the behavior of the knee adduction moment. However, no reports exist on the moment of internal rotation of the knee joint during hip-related peripheral fatigue tasks; only one study reported that the moment of internal rotation increased during jump-landing after prolonged running [20]. The internal rotation of the knee joint (internal rotation of the tibia) is similar to the posture at the time of ACL injury [1, 2], indicating that the risk of ACL injury in the transverse plane is increased in the case of HFT.

In terms of KFT, the average DKV changed after the fatigue tasks, except for the moment of hip internal rotation, with an increase in internal rotation in the knee joint and a decrease in hip adduction/internal rotation angle and knee adduction. Additionally, the adduction/internal rotation moment of both hip and knee joints decreased. Quadriceps and hamstrings fatigue significantly increased the external rotation angle and internal rotation moment of the knee [8]. Similarly, the increase in quadriceps activity was consistent with that reported in previous studies. Typically, EMG activity does not indicate an increase in the force generated by muscles; however, an increase in EMG amplitude is commonly observed in fatigue at submaximal contractions [21]. Therefore, as peak knee extension torque in the fatigue task was significantly reduced resulting in the fatigue of the quadriceps, we attributed the increase in muscle activity to fatigue.

No significant differences were observed in the peak amplitude of gluteus medius. Previously, studies have reported a minimal effect of hip abductor muscle fatigue on the EMG of the gluteus medius during landing motion [7, 22], indicating that hip abductor muscle fatigue does not affect the peak amplitude during landing. However, other effects occur on lower-extremity EMG, such as increased activity in the tibialis anterior muscle [23]. Therefore, gluteus medius fatigue compensates for the fatigue in the other muscles, such as the quadriceps, as mentioned earlier.

This study verified that HFT and KFT exhibit different effects on the biomechanics of lower limbs after a fatigue task, with KFT affecting the mean value during 100 ms after IC and HFT affecting the DKV in the peak vGRF. A few previous studies have examined the differences in exercise intensity based on the comparison of fatigue tasks [8, 9]; however, the present study is novel because two different fatigue tasks were performed by the same participants. Typically, an ACL injury is caused by rapid knee abduction and internal rotation within 40 ms of landing [1, 2], and ACL strain is the highest at peak vGRF during cutting motion [24]. Therefore, movement analyses of ACL injury are reported at a certain point during the early landing phase. Conversely, an ACL-reconstructed knee is known to be different from that of a healthy knee during single-leg hop landing after 60 ms from IC [25]; therefore, the time course of lower-limb biomechanics after landing should also be considered. In this study, we clarified the differences between average DKV during landing and DKV at peak vGRF in different fatigue tasks, which can aid the detection of parameters for preventing ACL injury or re-injury.

Certain limitations were observed in this study. First, the performed analysis was a descriptive laboratory study and only male participants were considered. However, hip abductor muscle fatigue with respect to the biomechanical effects of the fatigue task significantly reduces the hip abduction angle in females as well [7]. Additionally, no significant differences have been observed in any kinematics and kinetics after performing only the hip internal and external rotator exercises in females [26]. Furthermore, healthy male participants are known to exhibit a higher knee adduction moment at landing than female participants [27], and the differences in knee abduction moment reduce even after the fatigue task depending on the gender [28]. Therefore, the knee adduction during post-fatigue being higher than that during pre-fatigue in this study could be attributed to the fact that only male participants were targeted. Second, the EMG analysis was not synchronized with motion capture systems, and the definition of IC was inaccurate. Third, the target muscle was limited to the gluteus medius and quadriceps, and it was difficult to clarify the influence of other muscles around the hip joint and understand the compensation in the muscles around the knee joint. Studies have reported that gluteus medius activity increases at the expense of hip extensors [29]. As this aspect was not considered in the present study, the overactivity in the quadriceps did not cause changes in the biomechanics of the lower extremity during HFT. Finally, the types of sports and competition levels were not unified because all participants were healthy males. A study reported that basketball players exhibit a higher knee valgus angle than soccer players [30]. Therefore, clarifying the difference in the degree of fatigue within the same sports activity is crucial. We did not perform measurements related to bone morphology or alignment of the participants. Previous studies have reported that participants with valgus experiential have shorter vertical-jump distances [31]; however, it has not been possible to prove that participants with knee valgus are more affected by fatigue. In addition, the subjects of this study were young, and it cannot be applied to elderly people whose muscle strength decreases after a fatiguing task [32].

Despite the aforementioned limitations, the study findings can form the scientific basis for existing ACL injury prevention programs [12]. Furthermore, regardless of the type of fatigue task, our analysis determined that peripheral fatigue tasks are likely to induce a risk for ACL injuries. In muscle strength training after ACL injury or ACL reconstruction, we often target only the area around the knee joint or evaluate muscle performance without considering factors such as fatigue. This study was conducted on healthy subjects with no history of orthopedic surgery in the lower extremities, and with knee and hip joint function. We demonstrated that even normal subjects adopted postures that put them at risk for ACL injury during both fatigue tasks. This suggests the need to evaluate the multi-joint and fatigue status of muscle-strengthening training after ACL injury and ACL reconstruction in clinical practice.

## Supporting information

**S1 Checklist. *PLOS ONE* clinical studies checklist.**
(DOCX)

## Acknowledgments

We would like to thank Editage (www.editage.com) for English language editing.

## Author Contributions

**Conceptualization:** Makoto Asaeda, Yukio Mikami.

**Data curation:** Makoto Asaeda.

**Formal analysis:** Makoto Asaeda, Tomoya Ohnishi, Hideyuki Ito, So Miyahara.

**Investigation:** Makoto Asaeda, Kazuhiko Hirata, Tomoya Ohnishi.

**Methodology:** Makoto Asaeda.

**Project administration:** Kazuhiko Hirata.

**Resources:** Yukio Mikami.

**Software:** Tomoya Ohnishi.

**Supervision:** Yukio Mikami.

**Visualization:** Makoto Asaeda.

**Writing – original draft:** Makoto Asaeda.

**Writing – review & editing:** Yukio Mikami.

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
