## [Decision Letter · Decision Letter 0]

17 Oct 2023

PONE-D-23-20581Differences in lower-limb biomechanics during single-leg landing considering two peripheral fatigue tasksPLOS ONE

Dear Dr. Asaeda,

Thank you for submitting your manuscript to PLOS ONE. After careful consideration, we feel that it has merit but does not fully meet PLOS ONE’s publication criteria as it currently stands. Therefore, we invite you to submit a revised version of the manuscript that addresses the points raised during the review process.

We look forward to receiving your revised manuscript.

Kind regards,

Giuseppe Messina

Academic Editor

PLOS ONE

Journal Requirements:

Did you know that depositing data in a repository is associated with up to a 25% citation advantage (https://doi.org/10.1371/journal.pone.0230416)? If you’ve not already done so, consider depositing your raw data in a repository to ensure your work is read, appreciated and cited by the largest possible audience. You’ll also earn an Accessible Data icon on your published paper if you deposit your data in any participating repository (https://plos.org/open-science/open-data/#accessible-data).

Reviewers' comments:

Reviewer's Responses to Questions

**Comments to the Author**

1. Is the manuscript technically sound, and do the data support the conclusions?

Reviewer #1: Yes

Reviewer #2: Yes

2. Has the statistical analysis been performed appropriately and rigorously? 

Reviewer #1: Yes

Reviewer #2: Yes

3. Have the authors made all data underlying the findings in their manuscript fully available?

Reviewer #1: Yes

Reviewer #2: Yes

4. Is the manuscript presented in an intelligible fashion and written in standard English?

Reviewer #1: Yes

Reviewer #2: Yes

5. Review Comments to the Author

Reviewer #1: INTRODUCTION

-The introduction is well written. However, for an exaustive background authors should better argue the concept of fatigue. In fact, they address the concept of peripheral fatigue and this is also reported on the title. Thus, in my opinion, they should provide existing knowledge on what is fatigue, what is the difference between central fatigue and peripheral fatigue, and, what is the fatigability. Below some references suggested:

Enoka RM. Mechanisms of muscle fatigue: Central factors and task dependency. J Electromyogr Kinesiol. 1995 Sep;5(3):141-9.

Enoka RM, Duchateau J. Translating Fatigue to Human Performance. Med Sci Sports Exerc. 2016 Nov;48(11):2228-2238.

METHODS

-Authors should report the study design used.

-Please report the type of sampling process used.

-(page 4 lines 77-80): Authors referred to a “single center” and to “a poster in a public space”. Please describe the methods in detail. What kind of center was it? Where was the poster put up?

-Authors stated that: “We recruited those (i) who could understand the study consent in writing; (ii) without a history of orthopedic surgery in the lower limb neuromuscular disease; (iii) without pain; and (iv) exhibiting a limited range of motion in the lower limb, which affected their jump landing”. How were points (ii), (iii), and (iv) verified?

-(page 4 lines 80-83): Although the authors reported inclusion criteria, they failed to report exclusion criteria for inclusion.

-Details about data collection and setting should be reported. For example, who and where were performed the landing tasks?

-Were the participants familiar with the task to be performed?

RESULTS

-(Page 8 lines 162-166): In the results, authors reported that they separated the sample in “who had an ankle fracture in the previous year (n = 1) or experienced pain during landing (n = 1)”. How this aspect was assessed? Moreover, they stated: “Additionally, we excluded one participant who had difficulty maintaining the posture when jumping and landing”. This was not an exclusion criterion. According to what criteria did they do it? Furthermore, they reported: “…and two participants whose EMG data were lost during the fatigue task”. What exactly do the authors mean by this statement?

-(Page 8 lines 169-170): “All participants presented recreational levels in terms of sports activity, and none performed at the competitive level”. Again, how was it measured? Furthermore, I don't think there was an inclusion/exclusion criterion in this regard.

-In table 1 authors report anthropometric data. However, in the methods section no procedure for measuring weight and height was described.

DISCUSSION

-Among limitations, authors should mention that they did not previously assess the knee morphology and this aspect can influence jumping tasks performances (please consider the following: Giustino V, et al. Effects of a Postural Exercise Program on Vertical Jump Height in Young Female Volleyball Players with Knee Valgus. Int J Environ Res Public Health. 2022 Mar 26;19(7):3953).

-Authors failed to report the practical implications of their study.

Reviewer #2: In this article authors investigated dynamic knee valgus occurring during landing after two different fatigue tasks. They found that fatigue tasks, by inducing dynamic knee valgus, were likely to induce a risk for anterior cruciate ligament injury

Overall the paper appears to be well written and its results are interesting

I have only minor comments:

Methods

Authors should explain why they have chosen a distance of 20 cm for the jump

How was the sample size of this study chosen?

I suggest to provide a figure illustrating a subject while he is performing the fatigue tasks

Discussion

Since the average age of subjects who participated to this research is very low, in my opinion, results cannot be generalized to older people. This point should be mentioned among limitations

6. PLOS authors have the option to publish the peer review history of their article (what does this mean?). If published, this will include your full peer review and any attached files.

Reviewer #1: No

Reviewer #2: **Yes: **Giuseppe Caminiti

---

## [Author Response · Author response to Decision Letter 0]

15 Nov 2023

Responses to the comments from Reviewer #1.

Thank you for your constructive comments. We have revised the manuscript and tables based on your comments. 

1. The introduction is well written. However, for an exaustive background authors should better argue the concept of fatigue. In fact, they address the concept of peripheral fatigue and this is also reported on the title. Thus, in my opinion, they should provide existing knowledge on what is fatigue, what is the difference between central fatigue and peripheral fatigue, and, what is the fatigability. Below some references suggested:

Enoka RM. Mechanisms of muscle fatigue: Central factors and task dependency. J Electromyogr Kinesiol. 1995 Sep;5(3):141-9.

Enoka RM, Duchateau J. Translating Fatigue to Human Performance. Med Sci Sports Exerc. 2016 Nov;48(11):2228-2238.

Response

Thank you for the valuable comments. Following your suggestion, we added a description of peripheral fatigue, and we also incorporated the references you suggested.

" The physiological processes that can cause fatigue are classically divided into two domains: the activation level of the muscle (central) and the other influences on contractile function (peripheral)　[5]. Fatigue is also classified as perceived or performance fatigability, and it is determined by a wide variety of factors. Peripheral fatigue includes neuromuscular function, metabolism, contractile apparatus, and contraction coupling in peripheral muscles, which is related to fatigue as a task dependency　[5,6]." (Page 3, Lines 49 to 54)

2. Authors should report the study design used.

Response

Thank you for this comment. As you suggested, we have added the details of study design. (Page 4, line 83)

3. Please report the type of sampling process used.

Response

Thank you for this comment. To the best of our knowledge, there were no reports comparing the landing motion of a single-leg drop jump before and after different fatigue-inducing tasks. Therefore, we utilized the knee joint flexion angle data from a study that analyzed the motion before and after isokinetic knee joint exercises in the same sample for our sample-size calculations.

4. Authors referred to a “single center” and to “a poster in a public space”. Please describe the methods in detail. What kind of center was it? Where was the poster put up?

Response

Thank you for this comment. We targeted a single campus and recruited participants by posting posters on the bulletin board at the campus entrance. We have described this in detail on Page 4, Line 84 and 85.

5. Authors stated that: “We recruited those (i) who could understand the study consent in writing; (ii) without a history of orthopedic surgery in the lower limb neuromuscular disease; (iii) without pain; and (iv) exhibiting a limited range of motion in the lower limb, which affected their jump landing”. How were points (ii), (iii), and (iv) verified?

Response

Thank you for this comment. We conducted a questionnaire among prospective participants to confirm their history of orthopedic surgery, as well as the presence or absence of pain related to lower-limb neuromuscular diseases. Furthermore, to assess individuals with limited lower-limb range of motion, joint angles were measured using a goniometer.

The above information has been included in the manuscript.

“A questionnaire was administered to prospective participants to confirm their history of orthopedic surgery related to lower-limb neuromuscular diseases and to determine the presence or absence of pain. Additionally, joint angles were measured using a goniometer to assess individuals with limited range of motion in the lower limb. The exclusion criteria included individuals with a history of orthopedic surgery or lower-limb pain and those with limited joint range of motion (i.e., less than 135° of flexion), maintaining the posture when jumping and landing,” (Page 5, Lines 90 to 96)

6. Although the authors reported inclusion criteria, they failed to report exclusion criteria for inclusion.

Response

Thank you for this comment. We have added the following section on Page 5, Lines 93 to 95 

“The exclusion criteria included individuals with a history of orthopedic surgery or lower-limb pain and those with limited joint range of motion (i.e., less than 135° of flexion).”

7. Details about data collection and setting should be reported. For example, who and where were performed the landing tasks?

Response

Thank you for this comment. We have added the following text to the manuscript:

"Data were collected in a secured room, ensuring minimal external influence during measurements, and the measurements were conducted by two individuals: MA and HI." (Page 6, Lines 112 to 114)

“Data analysis was performed at the data collection site by a single individual using a computer having no internet connectivity. The analyzed results were reviewed by two individuals, TO and SM, to ensure their validity.” (Page 7, Lines 160 to 161)

8. Were the participants familiar with the task to be performed?

Response

Thank you for this question. All participants had a history of competitive sports and experience with fatigue tasks and other jump landing movements. However, we did not have any prior experience with this specific jump landing motion. Therefore, we have included the following statement: “All participants had a history of competing in sports but lacked prior experience in performing the specific jump landing motion used in this study.” (Page 6, Lines 124 to 125)

9. In the results, authors reported that they separated the sample in “who had an ankle fracture in the previous year (n = 1) or experienced pain during landing (n = 1)”. How this aspect was assessed? Moreover, they stated: “Additionally, we excluded one participant who had difficulty maintaining the posture when jumping and landing”. This was not an exclusion criterion. According to what criteria did they do it? Furthermore, they reported: “…and two participants whose EMG data were lost during the fatigue task”. What exactly do the authors mean by this statement?

Response

Thank you for this comment. We have add following sentence:

“Before the fatigue task, participants practiced the landing task three times,” (Page 6, Line 122)

“from the questionnaire before the landing task” (Page 9, Lines 188 to 189)

“during landing in the practice phase before the fatigue task” (Page 9, Lines 189 to 190)

“because of dropping the electrodes for surface electromyography during the fatigue task” (Page 9, Lines 192 to 193)

10. “All participants presented recreational levels in terms of sports activity, and none performed at the competitive level”. Again, how was it measured? Furthermore, I don't think there was an inclusion/exclusion criterion in this regard.

Response

Thank you for this comment. In our questionnaire, we asked whether they participate in regular sports or in conferences. Therefore, we added the following text

“(v) carrying out regular sports activities” (Page 5, Lines 89 to 90)

“and not carrying out regular sporting activities, carrying out sporting activities at competition level (e.g. professional sports level)” (Page 5, Lines 96 to 97)

11. In table 1 authors report anthropometric data. However, in the methods section no procedure for measuring weight and height was described.

Response

Thank you for this comment. As you mentioned, we have added the following sentence. 

“Heights and weights of all participants were respectively measured using height scale (Yamato, Japan) and weight scale (Yamato, Japan), and BMI was calculated” (Page 5, Lines 97 to 99)

12. Among limitations, authors should mention that they did not previously assess the knee morphology and this aspect can influence jumping tasks performances (please consider the following: Giustino V, et al. Effects of a Postural Exercise Program on Vertical Jump Height in Young Female Volleyball Players with Knee Valgus. Int J Environ Res Public Health. 2022 Mar 26;19(7):3953).

Response

Thank you for this suggestion. As you mentioned, we have added the following sentence and references: 

“We did not perform measurements related to bone morphology or alignment of the participants. Previous studies have reported that participants with valgus experiential have shorter vertical-jump distances [31]; however, it has not been possible to prove that participants with knee valgus are more affected by fatigue.” (Page 15, Lines 319 to 322)

“30. Giustino V, Messina G, Patti A, Padua E, Zangla D, Drid P, et al. Effects of a Postural Exercise Program on Vertical Jump Height in Young Female Volleyball Players with Knee Valgus. Int J Environ Res Public Health. 2022;19: 3953. doi:10.3390/ijerph19073953”

13. Authors failed to report the practical implications of their study.

Response

Thank you for this comment. We have added the following text: 

“In muscle strength training after ACL injury or ACL reconstruction, we often target only the area around the knee joint or evaluate muscle performance without considering factors such as fatigue. This study was conducted on healthy subjects with no history of orthopedic surgery in the lower extremities, and with knee and hip joint function. We demonstrated that even normal subjects adopted postures that put them at risk for ACL injury during both fatigue tasks. This suggests the need to evaluate the multi-joint and fatigue status of muscle-strengthening training after ACL injury and ACL reconstruction in clinical practice.” (Page 15 , Lines 328 to 334) 

 

Responses to the comments from Reviewer #2.

Thank you for the constructive comments. We have revised the manuscript and tables based on your comments.

1. Authors should explain why they have chosen a distance of 20 cm for the jump. How was the sample size of this study chosen?

Response

There have been various studies investigating jump landing behavior. For this study, we referred to the following previous research. Therefore, we have added the following sentence on Page 6, Lines 115 to 116. 

“this was based on a previous study [13]”

Additionally, to the best of our knowledge, there were no reports comparing the landing motion of a single-leg drop jump before and after different fatigue-inducing tasks. Therefore, we utilized the knee joint flexion angle data from a study that analyzed the motion before and after isokinetic knee joint exercises in the same sample for our sample-size calculations. 

2. I suggest to provide a figure illustrating a subject while he is performing the fatigue tasks

Response

Thank you for this comment. We have added Figure 1 during the fatigue task 

3. Since the average age of subjects who participated to this research is very low, in my opinion, results cannot be generalized to older people. This point should be mentioned among limitations

Response

Thank you for this comment. As you suggested, we have added the following sentence:

 “In addition, the subjects of this study were young, and it cannot be applied to elderly people whose muscle strength decreases after a fatiguing task [32].” (Page 15, Lines 322 to 324)

---

## [Decision Letter · Decision Letter 1]

16 Jan 2024

Differences in lower-limb biomechanics during single-leg landing considering two peripheral fatigue tasks

PONE-D-23-20581R1

Dear Dr. Asaeda,

We’re pleased to inform you that your manuscript has been judged scientifically suitable for publication and will be formally accepted for publication once it meets all outstanding technical requirements.

Kind regards,

Emiliano Cè

Academic Editor

PLOS ONE

Additional Editor Comments (optional):

Reviewers' comments:

Reviewer's Responses to Questions

**Comments to the Author**

1. If the authors have adequately addressed your comments raised in a previous round of review and you feel that this manuscript is now acceptable for publication, you may indicate that here to bypass the “Comments to the Author” section, enter your conflict of interest statement in the “Confidential to Editor” section, and submit your "Accept" recommendation.

Reviewer #1: All comments have been addressed

2. Is the manuscript technically sound, and do the data support the conclusions?

Reviewer #1: (No Response)

3. Has the statistical analysis been performed appropriately and rigorously? 

Reviewer #1: (No Response)

4. Have the authors made all data underlying the findings in their manuscript fully available?

Reviewer #1: (No Response)

5. Is the manuscript presented in an intelligible fashion and written in standard English?

Reviewer #1: (No Response)

6. Review Comments to the Author

Reviewer #1: (No Response)

7. PLOS authors have the option to publish the peer review history of their article (what does this mean?). If published, this will include your full peer review and any attached files.

Reviewer #1: No

---

## [Editor Report · Acceptance letter]

1 Apr 2024

PONE-D-23-20581R1 

PLOS ONE

Dear Dr. Asaeda, 

I'm pleased to inform you that your manuscript has been deemed suitable for publication in PLOS ONE. Congratulations! Your manuscript is now being handed over to our production team.

Kind regards, 

on behalf of

Prof. Emiliano Cè 

Academic Editor

PLOS ONE